# Measuring the Contribution of Multiple Model Representations in Detecting Adversarial Instances

**Daniel Steinberg,**[1] **Paul Munro**[2]

[1] Intelligent Systems Program, University of Pittsburgh
[2] School of Computing and Information, University of Pittsburgh
das178@pitt.edu, pwm@pitt.edu

## Abstract

Deep learning models have been used for a wide variety of tasks. They are prevalent in computer vision, natural language processing, speech recognition, and other areas. While these models have worked well under many scenarios, it has been shown that they are vulnerable to adversarial attacks. This has led to a proliferation of research into ways that such attacks could be identified and/or defended against. Our goal is to explore the contribution that can be attributed to using multiple underlying models for the purpose of adversarial instance detection. Our paper describes two approaches that incorporate representations from multiple models for detecting adversarial examples. We devise controlled experiments for measuring the detection impact of incrementally utilizing additional models. For many of the scenarios we consider, the results show that performance increases with the number of underlying models used for extracting representations.

Code is available at https://github.com/dstein64/multi-adv-detect.

## 1 Introduction

Research on neural networks has progressed for many decades, from early work modeling neural activity (McCulloch and Pitts 1943) to the more recent rise of deep learning (Bengio, Lecun, and Hinton 2021). Notable applications include image classification (Krizhevsky, Sutskever, and Hinton 2012), image generation (Goodfellow et al. 2014), image translation (Isola et al. 2017), and many others (Dargan et al. 2020). Along with the demonstrated success it has also been shown that carefully crafted adversarial instances—which appear as normal images to humans—can be used to deceive deep learning models (Szegedy et al. 2014), resulting in incorrect output. The discovery of adversarial instances has led to a broad range of related research including 1) the development of new attacks, 2) the characterization of attack properties, and 3) defense techniques. Akhtar and Mian present a comprehensive survey on the threat of adversarial attacks to deep learning systems used for computer vision.

Two general approaches—discussed further in Section 6—that have been proposed for defending against adversarial attacks include 1) the usage of model ensembling and 2) the incorporation of hidden layer representations as discriminative features for identifying perturbed data. Building on these ideas, we explore the performance implications that can be attributed to using representations from multiple models for the purpose of adversarial instance detection.

**Our Contribution** In Section 3 we present two approaches that use neural network representations as features for an adversarial detector. For each technique we devise a treatment and control variant in order to measure the impact of using multiple networks for extracting representations. Our controlled experiments in Section 4 measure the effect of using multiple models. For many of the scenarios we consider, detection performance increased as a function of the underlying model count.

## 2 Preliminaries

Our research incorporates $l$-layer feedforward neural networks, functions $h : \mathcal{X} \to \mathcal{Y}$ that map input $x \in \mathcal{X}$ to output $\hat{y} \in \mathcal{Y}$ through linear preactivation functions $f_i$ and nonlinear activation functions $\phi_i$.

$$\hat{y} = h(x) = \phi_l \circ f_l \circ \phi_{l-1} \circ f_{l-1} \circ \ldots \circ \phi_1 \circ f_1(x)$$

The models we consider are classifiers, where the outputs are discrete labels. For input $x$ and its true class label $y$, let $J(x, y)$ denote the corresponding loss of a trained neural network. Our notation omits the dependence on model parameters $\theta$, for convenience.

### 2.1 Adversarial Attacks

Consider input $x$ that is correctly classified by neural network $h$. For an untargeted adversarial attack, the adversary tries to devise a small additive perturbation $\Delta x$ such that adversarial input $x^{adv} = x + \Delta x$ changes the classifier's output (i.e., $h(x) \neq h(x^{adv})$). For a targeted attack, a desired value for $h(x^{adv})$ is an added objective. In both cases, the $L_p$ norm of $\Delta x$ is typically constrained to be less than some threshold $\epsilon$. Different threat models—white-box, grey-box, and black-box—correspond to varying levels of knowledge that the adversary has about the model being used, its parameters, and its possible defense.

The adversary's objective can be expressed as an optimization problem. For example, the following constrained maximization of the loss function is one way of formulating how an adversary could generate an untargeted adversarial

input $x^{adv}$.

$$\Delta x = \underset{\delta}{\operatorname{argmax}} \ J(x + \delta, y)$$
$$\text{subject to } \|\delta\|_p \leq \epsilon$$
$$x + \delta \in \mathcal{X}$$

There are various ways to generate attacks. Under many formulations it's challenging to devise an exact computation of $\Delta x$ that optimizes the objective function. An approximation is often employed.

**Fast Gradient Sign Method (FGSM)** (Goodfellow, Shlens, and Szegedy 2015) generates an adversarial perturbation $\Delta x = \epsilon \cdot \operatorname{sign}(\nabla_x J(x, y))$, which is the approximate direction of the loss function gradient. The $\operatorname{sign}$ function bounds its input to an $L_\infty$ norm of 1, which is scaled by $\epsilon$.

**Basic Iterative Method (BIM)** (Kurakin, Goodfellow, and Bengio 2017) iteratively applies FGSM, whereby $x_t^{adv} = x_{t-1}^{adv} + \alpha \cdot \operatorname{sign}(\nabla_x J(x_{t-1}^{adv}, y))$ for each step, starting with $x_0^{adv} = x$. The $L_\infty$ norm is bounded by $\alpha$ on each iteration and by $t \cdot \alpha$ after $t$ iterations. $x_t^{adv}$ can be clipped after each iteration in a way that constrains the final $x^{adv}$ to an $\epsilon$-ball of $x$.

**Carlini & Wagner (CW)** (Carlini and Wagner 2017) generates an adversarial perturbation via gradient descent to solve $\Delta x = \operatorname{argmin}_\delta(\|\delta\|_p + c \cdot f(x + \delta))$ subject to a box constraint on $x + \delta$. $f$ is a function for which $f(x + \delta) \leq 0$ if and only if the target classifier is successfully attacked. Experimentation yielded the most effective $f$—for targeted attacks—of those considered. $c$ is a positive constant that can be found with binary search, a strategy that worked well empirically. Clipping or a change of variables can be used to accommodate the box constraint.

## 2.2 Ensembling

Our research draws inspiration from ensembling, the combination of multiple models to improve performance relative to the component models themselves. There are various ways of combining models. An approach that is widely used in deep learning averages outputs from an assortment of neural networks; each network having the same architecture, trained from a differing set of randomly initialized weights.

# 3 Method

To detect adversarial instances, we use hidden layer representations—from *representation models*—as inputs to adversarial *detection models*. For our experiments in Section 4, the representation models are convolutional neural networks that are independently trained for the same classification task, initialized with different weights. Representations are extracted from the penultimate layers of the trained networks. The method we describe in this section is more general, as various approaches could be used for preparing representation models. For example, each representation model could be an independently trained autoencoder—as opposed to a classifier—with representations for each model extracted from arbitrary hidden layers. Additionally,

it's not necessary that each of the models—used for extracting representations—has the same architecture.

We devise two broad techniques—*model-wise* and *unit-wise*—for extracting representations and detecting adversarial instances. These approaches each have two formulations, a *treatment* that incorporates multiple representation models and a *control* that uses a single representation model. For each technique, the functional form of the detection step is the same across treatment and control. This serves our objective of measuring the contribution of incrementally incorporating multiple representation models, as the control makes it possible to check whether gains are coming from some aspect other than the incorporation of multiple representation models.

The illustrations in this section are best viewed in color.

## 3.1 Model-Wise Detection

With $N$ representation models, model-wise detection uses a set of representations from each underlying model as separate input to $N$ corresponding detection models that each outputs an adversarial score. These scores, which we interpret as estimated probabilities, are then averaged to give an ensemble adversarial probability estimate. A baseline—holding fixed the number of detectors—uses a single representation model as a repeated input to multiple detection models. The steps of both approaches are outlined below.

**Model-Wise Treatment**

**Step 1** Extract representations for input $x$ from $N$ representation models.

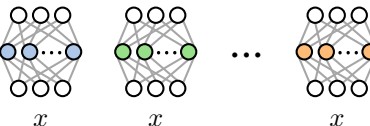

**Step 2** Pass the *Step 1* representations through $N$ corresponding detection models that each output adversarial probability (denoted $P_i$ for model $i$).

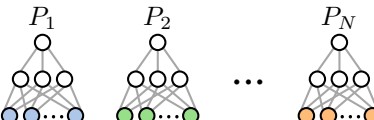

**Step 3** Calculate adversarial probability $P$ as the average of *Step 2* adversarial probabilities.

$$P = \frac{1}{N} \sum_{i=1}^{N} P_i$$

**Model-Wise Control**

**Step 1** Extract representations for input $x$ from a single representation model.

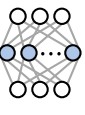

**Step 2** Pass the *Step 1* representations through $N$ detection models that each outputs adversarial probability (denoted $P_i$ for model $i$).

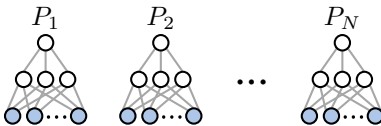

**Step 3** Calculate adversarial probability $P$ as the average of *Step 2* adversarial probabilities.

$$P = \frac{1}{N} \sum_{i=1}^{N} P_i$$

## 3.2 Unit-Wise Detection

With $N$ representation models, model-wise detection incorporates a single representation from each underlying model to form an $N$-dimensional array of features as input to a single detection model. A baseline—holding fixed the number of features for the detector—uses a set of units from a single representation model to form an input array for a detection model. The steps of both approaches are outlined below.

**Unit-Wise Treatment**

**Step 1** Extract a single representation for input $x$ from $N$ representation models. There is no requirement on which unit is selected nor whether there is any correspondence between which unit is selected from each model.

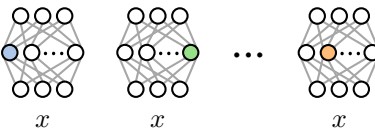

**Step 2** Pass the $N$-dimensional array of *Step 1* representations through an adversarial detection model that outputs adversarial probability $P$.

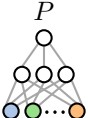

**Unit-Wise Control**

**Step 1** Extract $N$ units from the representations for input $x$ from a single representation model. In the illustration that follows, the count of extracted representation units, $N$, matches the total number of units available. It's also possible for $N$ to be smaller than the quantity available.

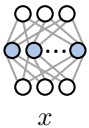

**Step 2** Pass *Step 1* representations through an adversarial detection model that outputs adversarial probability $P$.

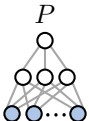

## 3.3 Measuring the Contribution from Multiple Models

We are interested in measuring the contribution of multiple models for detecting adversarial instances. For both the model-wise and unit-wise detection techniques, the contribution of multiple models can be evaluated by inspecting the change in treatment performance when incrementing the number of representation models, $N$. The changes should be considered relative to the control performance, to check whether any differences are coming from some aspect other than the incorporation of multiple representation models.

# 4 Experiments

## 4.1 Experimental Settings

We conducted experiments using the CIFAR-10 dataset (Krizhevsky 2009), which is comprised of 60,000 $32 \times 32$ RGB images across 10 classes. The dataset, as received, was already split into 50,000 training images and 10,000 test images. We trained one neural network classifier that served as the target for generating adversarial attacks. We trained 1,024 additional neural network classifiers to be used as representation models—with representations extracted from the 512-dimensional penultimate layer of each network. A different randomization seed was used for initializing the weights of the 1,025 networks. Each network had the same—18-layer, 11,173,962-parameter—ResNet-inspired architecture, with filter counts and depth matching the kuangliu ResNet-18 architecture.[1] Pixel values of input images were scaled by $1/255$ to be between 0 and 1. The networks were trained for 100 epochs using an Adam optimizer (Kingma and Ba 2014), with random horizontal flipping and random crop sampling on images padded with 4 pixels per edge. The model for attack generation had 91.95% accuracy on the test dataset. The average test accuracy across the 1,024 additional networks was 92.22% with sample standard deviation of 0.34%.

**Adversarial Attacks** Untargeted adversarial perturbations were generated for the 9,195 images that were originally correctly classified by the attacked model. Attacks were conducted with FGSM, BIM, and CW, all using the `cleverhans` library (Papernot et al. 2018). After each attack, we clipped the perturbed images between 0 and 1 and quantized the pixel intensities to 256 discrete values. This way the perturbed instances could be represented in 24-bit RGB space.

For FGSM, we set $\epsilon = 3/255$ for a maximum perturbation of 3 intensity values (out of 255) for each pixel on the unnormalized data. Model accuracy on the attacked model—for the 9,195 perturbed images—was 21.13% (i.e., an attack success rate of 78.87%). Average accuracy on the 1,024 representation models was 61.69% (i.e., an attack transfer success rate of 38.31%) with sample standard deviation of 1.31%.

For BIM, we used 10 iterations with $\alpha = 1/255$ and maximum perturbation magnitude clipped to $\epsilon = 3/255$. This re-

---

[1]This differs from the ResNet-20 architecture used for CIFAR-10 in the original ResNet paper (He et al. 2016).

|  | Original | FGSM | BIM | CW |
|---|---|---|---|---|

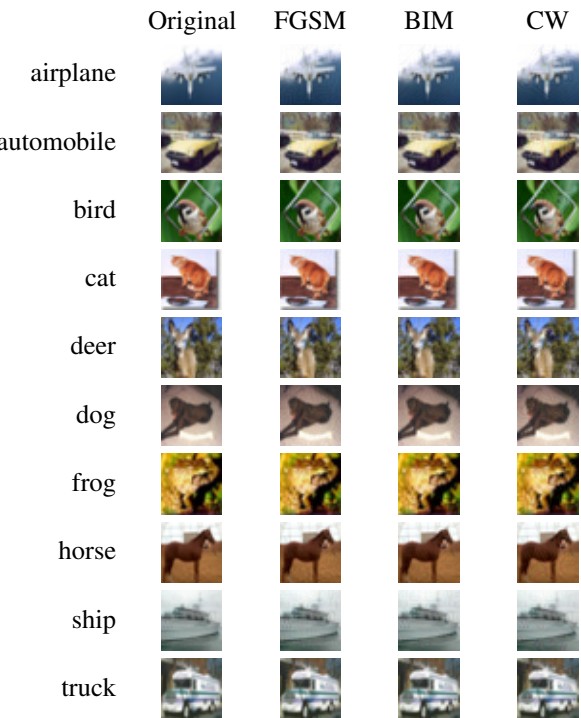

airplane
automobile
bird
cat
deer
dog
frog
horse
ship
truck

Figure 1: Example CIFAR-10 images after adversarial perturbation. The original image—in the leftmost column—is followed by three columns corresponding to FGSM, BIM, and CW attacks, respectively. Images were chosen randomly from the set of test images that were correctly classified without perturbation—the population of images for which attacks were generated.

sults in a maximum perturbation of 1 unnormalized intensity value per pixel on each step, with maximum perturbation after all steps clipped to 3. Accuracy after attack was 0.61% for the attacked model. Average accuracy on the 1,024 representation models was 41.09% with sample standard deviation of 2.64%.

For CW, we used an $L_2$ norm distance metric along with most default parameters—a learning rate of 0.005, 5 binary search steps, and 1,000 maximum iterations. We raised the confidence parameter[2] to 100 from its default of 0, which increases attack transferability. This makes our experiments more closely align with black-box and grey-box attack scenarios, where transferability would be an objective of an adversary. Accuracy after attack was 0.07% for the attacked model. Average accuracy on the 1,024 representation models was 5.86% with sample standard deviation of 1.72%.

Figure 1 shows examples of images that were perturbed for our experiments. These were chosen randomly from the 9,195 correctly classified test images—the population of images for which attacks were generated.

---

[2]Our description of CW in Section 2 does not discuss the $\kappa$ confidence parameter. See the CW paper (Carlini and Wagner 2017) for details.

**Adversarial Detectors** We use the 512-dimensional representation vectors extracted from the 1,024 representation models as inputs to model-wise and unit-wise adversarial detectors—both treatment and control configurations—as described in Section 3. All detection models are binary classification neural networks that have a 100-dimensional hidden layer with a rectified linear unit activation function. We did not tune hyperparameters, instead using the defaults as specified by the library we employed, scikit-learn (Pedregosa et al. 2011). Model-wise detectors differed in their randomly initialized weights.

To evaluate the contribution of multiple models, we run experiments that vary 1) the number of detection models used for model-wise detection, and 2) the number of units used for unit-wise detection. For the treatment experiments, the number of underlying representation models matches 1) the number of detection models for model-wise detection and 2) the number of units for unit-wise detection. For the control experiments, there is a single underlying representation model.

The number of units for the unit-wise control models was limited to 512, based on the dimensionality of the penultimate layer representations. The number of units for the unit-wise treatment was extended beyond this since its limit is based on the number of representation models, for which we had more than 512. One way to incorporate more units into the unit-wise control experiments would be to draw units from other network layers, but we have not explored that for this paper.

We are interested in the generalization capabilities of detectors trained with data from a specific attack. While the training datasets we constructed were each limited to a single attack algorithm, we separately tested each model using data attacked with each of the three algorithms—FGSM, BIM, and CW.

For training and evaluating each detection model, the dataset consisted of 1) the 9,125 images that were originally correctly classified by the attacked model, and 2) the 9,125 corresponding perturbed variants. Models were trained with 90% of the data and tested on the remaining 10%. Each original image and its paired adversarial counterpart were grouped, i.e., they were never separated such that one would be used for training and the other for testing.

We retained all 9,125 perturbed images and handled them the same (i.e., they were given the same class) for training and evaluation, including the instances that did not successfully deceive the attacked model. For BIM and CW, the consequence of this approach is presumably minor, since there were few unsuccessful attacks. For FGSM, which had a lower attack success rate, further work would be needed to 1) study the implications and/or 2) implement an alternative approach.

We conducted 100 trials for each combination of settings. For each trial, random sampling was used for 1) splitting data into training and test groups, 2) choosing representation models, and 3) choosing which representation units to use for the unit-wise experiments.

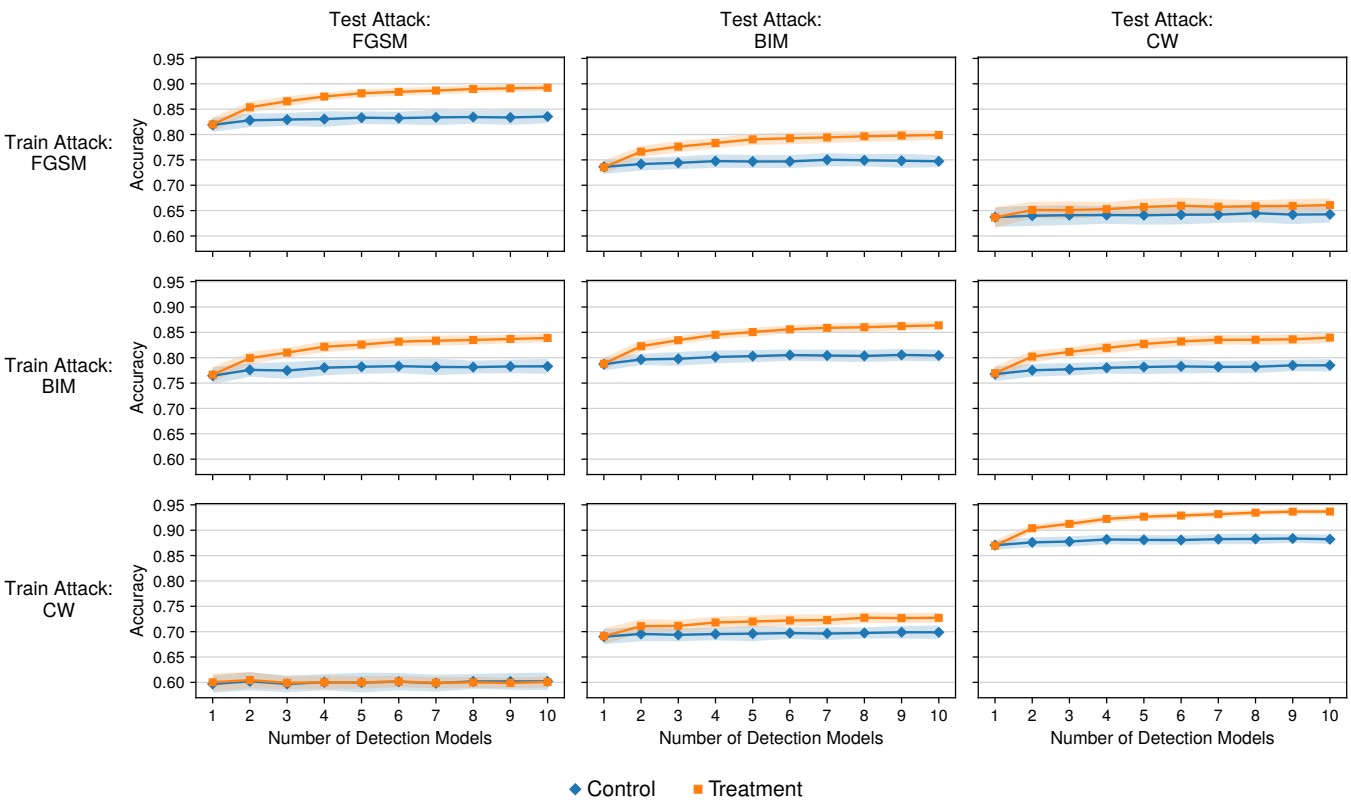

Figure 2: Average model-wise adversarial input detection accuracies, where each point is calculated across 100 trials. The sample standard deviations were added and subtracted from each sample mean to generate the shaded regions. The figure subplots each correspond to a specific attack used for the training data—as indicated by the leftmost labels—and a specific attack used for the test data—as indicated by the header labels. The endpoint values underlying the figure are provided in the appendix.

## 4.2 Hardware and Software

The experiments were conducted on a desktop computer running Ubuntu 21.04 with Python 3.9. The hardware includes an AMD Ryzen 9 3950X CPU, 64GB of memory, and an NVIDIA TITAN RTX GPU with 24GB of memory. The GPU was used for training the CIFAR-10 classifiers and generating adversarial attacks.

The code for the experiments is available at https://github.com/dstein64/multi-adv-detect.

## 4.3 Results

**Model-Wise** Figure 2 shows average model-wise adversarial input detection accuracies—calculated from 100 trials—plotted across the number of detection models. The subplots represent different combinations of training data attacks and test data attacks. The endpoint values underlying the figure are provided in the appendix.

**Unit-Wise** Figure 3 shows average unit-wise adversarial input detection accuracies—calculated from 100 trials—plotted across the number of units. The subplots represent different combinations of training data attacks and test data attacks. The endpoint values underlying the figure are provided in the appendix.

## 5 Discussion

Although subtle, for most scenarios the model-wise control experiments show an upward trend in accuracy as a function of the number of detection models. This is presumably an ensembling effect where there are benefits from combining multiple detection models even when they're each trained on the same features. The model-wise treatment experiments tend to outpace the corresponding controls, highlighting the benefit realized when the ensemble utilizes representations from distinct models.

The increasing accuracy for the unit-wise control experiments—as a function of the number of units—is more discernible than for the corresponding model-wise control experiments (the latter being a function of the number of models). The unit-wise gains are from having more units, and thus more information, as discriminative features for detecting adversarial instances. In most scenarios the treatment experiments—which draw units from distinct representation models—have higher performance than the corresponding controls. An apparent additional benefit is being able to incorporate more units when drawing from multiple models, not limited by the quantity of eligible units in a single model. However, drawing units from multiple models also comes at a practical cost, as it requires more computation relative to

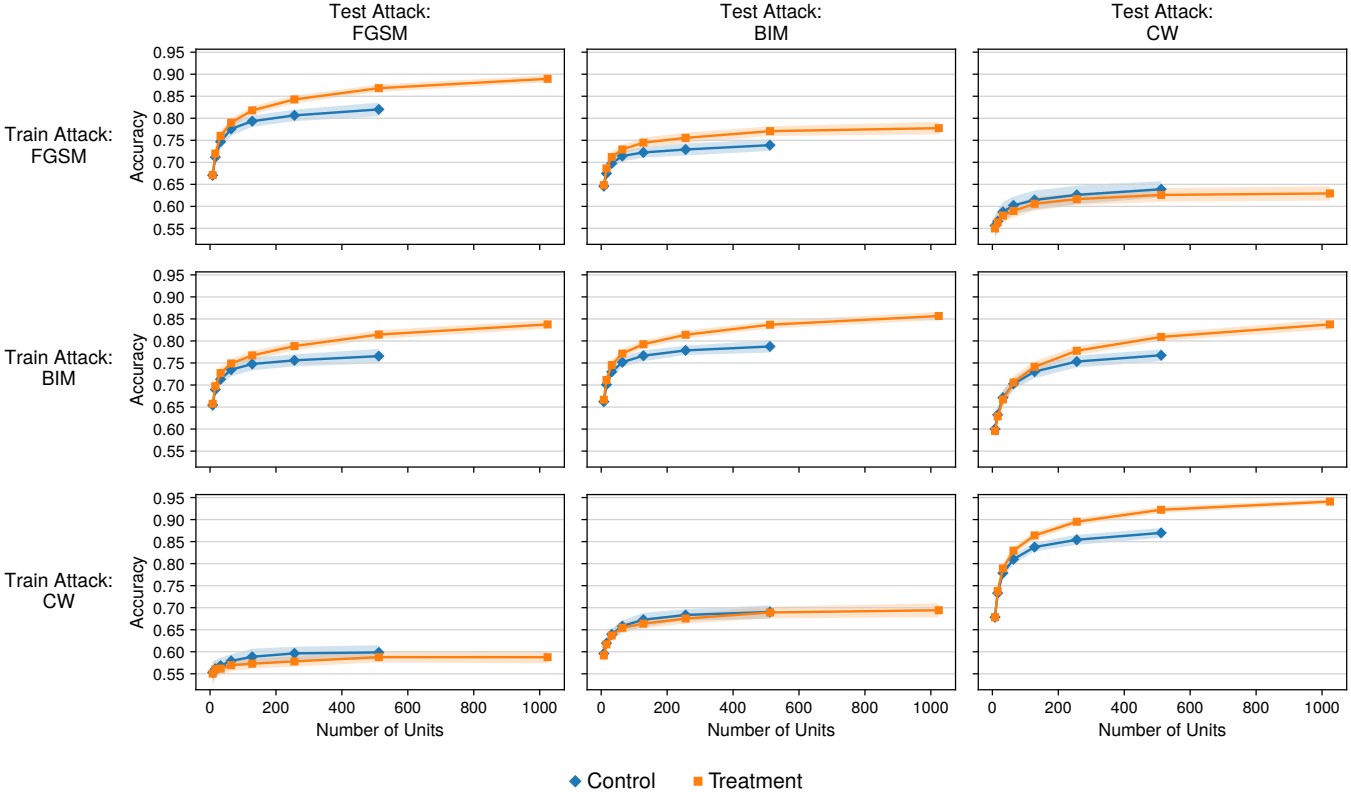

Figure 3: Average unit-wise adversarial input detection accuracies, where each point is calculated across 100 trials. The sample standard deviations were added and subtracted from each sample mean to generate the shaded regions. The figure subplots each correspond to a specific attack used for the training data—as indicated by the leftmost labels—and a specific attack used for the test data—as indicated by the header labels. The endpoint values underlying the figure are provided in the appendix.

drawing from a single model.

As expected, detectors trained with data from a specific attack perform best when tested with data from the same attack. Interestingly, detectors trained with BIM attack data appear to generalize better relative to detectors trained with FGSM or CW attack data. This may be related to the hyperparameters we used for each of the attacks, as opposed to being something representative of BIM more generally.

# 6 Related Work

We are aware of two general research areas that are related to what we've explored in this paper. The approaches include 1) the incorporation of ensembling for adversarial defense, and 2) the usage of hidden layer representations for detecting adversarial instances.

## 6.1 Ensembling-Based Adversarial Defense

Combining machine learning models is the hallmark of ensembling. For our work, we trained detection models that process representations extracted from multiple independently trained models. For model-wise detection, we averaged detection outputs across multiple models. Existing research has explored ensembling techniques in the context of defending against adversarial attacks (Liu et al. 2019). Bagnall, Bunescu, and Stewart train an ensemble—to be used

for the original task, classification, and also for adversarial detection—such that the underlying models agree on clean samples and disagree on perturbed examples. The *adaptive diversity promoting regularizer* (Pang et al. 2019) was developed to increase model diversity—and decrease attack transferability—among the members of an ensemble. Abbasi et al. devise a way to train ensemble *specialists* and merge their predictions—to mitigate the risk of adversarial examples.

## 6.2 Attack Detection from Representations

For our research we've extracted representations from independently trained classifiers to be used as features for adversarial example detectors. Hidden layer representations have been utilized in various other work on adversarial instance detection. Neural network invariant checking (Ma et al. 2019) detects adversarial samples based on whether internal activations conflict with invariants learned from nonadversarial data. Wójcik et al. use hidden layer activations to train autoencoders whose own hidden layer activations—along with reconstruction error—are used as features for attack detection. Li and Li develop a cascade classifier that incrementally incorporates statistics calculated on convolutional layer activations. At each stage, the instance is either classified as non-adversarial or passed along to the next

Table 1: Average unit-wise adversarial input detection accuracies plus/minus sample standard deviations, calculated across 100 trials for each datum. These are a subset of values used to generate Figure 2.

| Train Attack | Number of Detection Models | Test Attack | | | | | |
| | | FGSM | | BIM | | CW | |
| | | Control | Treatment | Control | Treatment | Control | Treatment |
|---|---|---|---|---|---|---|---|
| FGSM | 1 | $0.819 \pm 0.014$ | $0.820 \pm 0.014$ | $0.736 \pm 0.014$ | $0.735 \pm 0.014$ | $0.638 \pm 0.019$ | $0.637 \pm 0.020$ |
| | 10 | $0.836 \pm 0.013$ | $0.892 \pm 0.006$ | $0.747 \pm 0.012$ | $0.799 \pm 0.009$ | $0.643 \pm 0.017$ | $0.661 \pm 0.013$ |
| BIM | 1 | $0.765 \pm 0.017$ | $0.766 \pm 0.015$ | $0.788 \pm 0.013$ | $0.788 \pm 0.012$ | $0.767 \pm 0.014$ | $0.770 \pm 0.014$ |
| | 10 | $0.783 \pm 0.015$ | $0.839 \pm 0.009$ | $0.805 \pm 0.012$ | $0.864 \pm 0.008$ | $0.785 \pm 0.012$ | $0.840 \pm 0.010$ |
| CW | 1 | $0.597 \pm 0.017$ | $0.600 \pm 0.017$ | $0.690 \pm 0.015$ | $0.691 \pm 0.016$ | $0.870 \pm 0.009$ | $0.870 \pm 0.010$ |
| | 10 | $0.602 \pm 0.018$ | $0.601 \pm 0.011$ | $0.699 \pm 0.014$ | $0.727 \pm 0.010$ | $0.883 \pm 0.009$ | $0.937 \pm 0.005$ |

Table 2: Average unit-wise adversarial input detection accuracies plus/minus sample standard deviations, calculated across 100 trials for each datum. These are a subset of values used to generate Figure 3.

| Train Attack | Number of Units | Test Attack | | | | | |
| | | FGSM | | BIM | | CW | |
| | | Control | Treatment | Control | Treatment | Control | Treatment |
|---|---|---|---|---|---|---|---|
| FGSM | 8 | $0.671 \pm 0.014$ | $0.671 \pm 0.013$ | $0.646 \pm 0.012$ | $0.648 \pm 0.014$ | $0.556 \pm 0.024$ | $0.550 \pm 0.026$ |
| | 512 | $0.820 \pm 0.016$ | $0.868 \pm 0.008$ | $0.739 \pm 0.013$ | $0.771 \pm 0.011$ | $0.639 \pm 0.019$ | $0.626 \pm 0.016$ |
| | 1,024 | – | $0.890 \pm 0.008$ | – | $0.778 \pm 0.014$ | – | $0.629 \pm 0.016$ |
| BIM | 8 | $0.654 \pm 0.013$ | $0.657 \pm 0.014$ | $0.662 \pm 0.012$ | $0.667 \pm 0.013$ | $0.600 \pm 0.019$ | $0.596 \pm 0.020$ |
| | 512 | $0.766 \pm 0.017$ | $0.815 \pm 0.010$ | $0.787 \pm 0.014$ | $0.837 \pm 0.009$ | $0.768 \pm 0.013$ | $0.809 \pm 0.009$ |
| | 1,024 | – | $0.838 \pm 0.010$ | – | $0.857 \pm 0.010$ | – | $0.838 \pm 0.011$ |
| CW | 8 | $0.553 \pm 0.024$ | $0.550 \pm 0.026$ | $0.596 \pm 0.018$ | $0.592 \pm 0.019$ | $0.679 \pm 0.015$ | $0.678 \pm 0.017$ |
| | 512 | $0.599 \pm 0.016$ | $0.588 \pm 0.012$ | $0.690 \pm 0.015$ | $0.689 \pm 0.013$ | $0.870 \pm 0.011$ | $0.922 \pm 0.007$ |
| | 1,024 | – | $0.588 \pm 0.014$ | – | $0.694 \pm 0.016$ | – | $0.941 \pm 0.006$ |

stage of the cascade that integrates features computed from an additional convolutional layer. In addition to the methods summarized above, detection techniques have also been developed that 1) model the relative-positioned dynamics of representations passing through a neural network (Carrara et al. 2019), 2) use hidden layer activations as features for a $k$-nearest neighbor classifier (Carrara et al. 2017), and 3) process the hidden layer units that were determined to be relevant for the classes of interest (Granda, Tuytelaars, and Oramas 2020).

## 7 Conclusion and Future Work

We presented two approaches for adversarial instance detection—model-wise and unit-wise—that incorporate the representations from multiple models. Using those two approaches, we devised controlled experiments comprised of treatments and controls, for measuring the contribution of multiple model representations in detecting adversarial instances. For many of the scenarios we considered, experiments showed that detection performance increased with the number of underlying models used for extracting representations.

The research leaves open various avenues for future work.

- For our experiments, we trained 1,024 neural network representation models, whose diversity arises from using a different randomization seed for each. Perhaps other methods for imposing diversity would impact the performance of the detectors that depend on those models.

- It would be interesting to explore how existing adversarial defenses fare when extended to use multiple underlying models.

- Although we evaluated detectors across different attack algorithms, we always used data from a single attack for the purpose of training. Future research could investigate the effect of training with data from multiple attacks and/or varying hyperparameter settings for a specific attack.

- Our focus was on measuring the incremental gains of detecting attacks when incorporating multiple representation models. Further work could perform a thorough defense evaluation under more challenging threat models.

## Appendix

The endpoint values underlying Figure 2 are included in Table 1. The endpoint values underlying Figure 3 are included in Table 2.

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
