# OpenReview forum: "Measuring the Contribution of Multiple Model Representations in Detecting Adversarial Instances"
_AAAI.org/2022/Workshop/AdvML — AAAI-22 AdvML Workshop LongPaper_

### Official Review · Reviewer_UUsc · 2021-11-27
**Interesting idea of ensembling models to detect adversarial instances**

**Rating:** 7
**Confidence:** 4

**Review:**

This paper proposed to detect adversarial instances by ensembling the deep representations of multiple models in two ways ---- model-wise or unit-wise. The experiments is conducted on CIFAR-10 with the attack methods of FGSM, BIM and CW. The results show that with the incremental number of ensemble models or units, the detection accuracy will increase.

Strengths:
* The paper is well-written and the figures clearly convey the main ideas.
* The idea of incorporating the representations from multiple models to detect adversarial instances is straightforward yet effective.

Weaknesses:
* Comparison with existing adversarial example detection methods should be given.
* Experiments on other datasets can be further conducted.

---

### Official Review · Reviewer_tpsd · 2021-11-29
**An adversarial detector based on multiple model representations**

**Rating:** 7
**Confidence:** 4

**Review:**

This paper proposes two approaches using multiple model representations to detect adversarial examples. The authors conducted ablation studies to verify the contribution of the number of underlying models. The major weakness is that comparisons with other adversarial detectors are lacking.

---

### Decision · Program_Chairs · 2021-12-01

**Decision:**

Accept (Long Paper)

**Comment:**

Both reviewers agree to accept this paper.